# The relationship between body mass index and neurologic outcomes in survivors of out-of-hospital cardiac arrest treated with targeted temperature management

**Hyo Jin Bang[1], Kyu Nam Park[1], Chun Song Youn[1], Han Joon Kim[1], Sang Hoon Oh[1], Jee Yong Lim[1], Hwan Song[1], Soo Hyun Kim[2], Won Jung Jeong[3], Hyo Joon Kim[1]\*, on behalf of the Korean Hypothermia Network Investigator[¶]**

1 Department of Emergency Medicine, Seoul St. Mary Hospital, College of Medicine, College of Medicine, The Catholic University of Korea, Seoul, South Korea, 2 Department of Emergency Medicine, Eunpyeong St. Mary Hospital, College of Medicine, The Catholic University of Korea, Seoul, South Korea, 3 Department of Emergency Medicine, St. Vincent's Hospital, The Catholic University of Korea College of Medicine, Suwon, South Korea

¶ Membership of Korean Hypothermia Network Investigators is provided in Acknowledgments section.
\* khjoon0110@gmail.com

**Data Availability Statement:** All relevant data are within the manuscript and its Supporting Information files.

## Abstract

### Background

The association of body mass index with outcome in patients treated with targeted temperature management (TTM) after out-of-hospital cardiac arrest (OHCA) is unclear. The purpose of this study was to examine the effect of body mass index (BMI) on neurological outcomes and mortality in resuscitated patients treated with TTM after OHCA.

### Methods

This multicenter, prospective, observational study was performed with data from 22 hospitals included in the Korean Hypothermia Network KORHN-PRO registry. Comatose adult patients treated with TTM after OHCA between October 2015 and December 2018 were enrolled. The BMI of each patient was calculated and classified according to the criteria of the World Health Organization (WHO). Each group was analyzed in terms of demographic characteristics and associations with six-month neurologic outcomes and mortality after cardiac arrest (CA).

### Results

Of 1,373 patients treated with TTM identified in the registry, 1,315 were included in this study. One hundred two patients were underweight (BMI <18.5 kg/m$^2$), 798 were normal weight (BMI 18.5–24.9 kg/m$^2$), 332 were overweight (BMI 25–29.9 kg/m$^2$), and 73 were obese (BMI $\geq$ 30 kg/m$^2$). The higher BMI group had younger patients and a greater incidence of diabetes and hypertension. Six-month neurologic outcomes and mortality were not different among the BMI groups (p = 0.111, p = 0.234). Univariate and multivariate analyses

**Funding:** The author(s) received no specific funding for this work.

**Competing interests:** The authors have declared that no competing interests exist.

showed that BMI classification was not associated with six-month neurologic outcomes or mortality. In the subgroup analysis, the underweight group treated with TTM at 33˚C was associated with poor neurologic outcomes six months after CA (OR 2.090, 95% CI 1.010–4.325, p = 0.047), whereas the TTM at 36˚C group was not (OR 0.88, 95% CI 0.249–3.112, p = 0.843).

## Conclusions

BMI was not associated with six-month neurologic outcomes or mortality in patients surviving OHCA. However, in the subgroup analysis, underweight patients were associated with poor neurologic outcomes when treated with TTM at 33˚C.

## Introduction

The number of obese patients has been steadily increasing worldwide, and in the United States, the proportion increased from 30.5% in 1999–2000 to 42.4% in 2017–2018 [1]. Body mass index (BMI) is a simple tool that uses height and weight to measure obesity. As the number of obese patients increases, an increasing number of studies on the association between body mass index and various diseases has been performed, and the results are controversial. Obesity is known to be a major risk factor for chronic diseases, such as hypertension, type 2 diabetes mellitus, metabolic syndrome, myocardial infarction, and heart failure, which may increase the risk of cardiac arrest [2, 3]. However, the paradox of obesity is that obese patients with sepsis, which has high mortality rates, have better outcomes and survival [4, 5]. Another study found that overweight patients had the lowest mortality rate, and a lower BMI tended to increase all-cause mortality rates [6, 7].

Previous studies examining the relationship between out-of-hospital cardiac arrest (OHCA) outcomes and BMI had controversial results. One single-center, retrospective study showed that an overweight BMI was associated with lower six-month mortality and poor neurologic outcomes in OHCA patients treated with targeted temperature management (TTM) [8]. Another study showed that obesity is a risk factor for mortality in OHCA patients treated with therapeutic hypothermia [9, 10], and other studies showed that BMI was not associated with survival or good neurologic outcomes at hospital discharge [11, 12]. However, in all of the above studies, patients were treated at a target temperature of 33˚C, and neurologic outcomes were evaluated at hospital discharge.

In the current study, we aimed to evaluate whether BMI influences six-month neurologic outcomes in patients treated with TTM after cardiac arrest using a multicenter, prospective registry. Through subgroup analysis, we aimed to investigate differences in neurological outcomes according to BMI in the TTM 33 and TTM 36 groups.

## Methods

### Study setting and participation

This registry-based study was conducted prospectively with data from 22 centers included in the Korean Hypothermia Network, KORHN-PRO registry (NCT02827422). The study included an informed consent form approved by all participating hospitals, including the institutional review board (IRB) of Seoul St. Mary's Hospital (XC150IMI0081K). Between October 2015 and December 2018, comatose adult patients (≥ 18 years old) who were resuscitated

following OHCA and treated with TTM were registered in the KORHN-PRO registry. The registry excluded prearrest CPC 3 or 4; known disease resulting in death at 180 days; and cardiac arrest caused by trauma, stroke, or intracranial bleeding. The web-based registry consisted of patient information, resuscitation variables, TTM information, in-hospital treatment, outcomes and outcome prediction modalities. Each center received approval to participate by its Institutional Review Board. Written informed consent was obtained from all patients.

The TTM protocol of all centers included the same TTM parameters. The TTM regimen involved cooling from 32˚C to 36˚C for 12 to 24 h and controlled normothermia for 72 h after the return of spontaneous circulation. The participating hospitals treated the patients according to their own treatment protocols, and the TTM device and target temperature selection were based on individual preferences. For subgroup analysis, the patients were divided into two groups according to their intended regimen. Patients planning to be treated with a low temperature setting of 33–34˚C were considered the TTM 33 group, and those planning to be treated with a high temperature setting of 35–36˚C were considered the TTM 36 group. Neurologic outcomes were investigated by researchers at each hospital using multimodal tests.

## Data collection and definitions

We obtained data from the KORHN-PRO registry in the Utstein style. We included adult OHCA survivors who underwent TTM with a target temperature of 32–36˚C for 24 h and excluded patients 1) whose weight or height were missed and 2) who had no data on 6-month neurologic outcomes. The following information was obtained: 1) patients characteristics: sex, age, comorbidities (coronary heart disease, hypertension, diabetes mellitus, cardiovascular accident, lung disease and renal disease), height and weight; 2) resuscitation variables: arrest location, initial arrest rhythm, witness of collapse, bystander cardiopulmonary resuscitation (CPR), arrest etiology (cardiac or noncardiac) and anoxic time (collapse to ROSC time); 3) TTM: TTM methods, target temperature, induction time (time from start of TTM to the time of achieving the target temperature), rewarming time, time from ROSC to start of TTM, and the time from ROSC to target temperature achievement; and 4) in-hospital data: shock after ROSC (systolic blood pressure < 90 mmHg for > 30 minutes, or the need for supportive measures to maintain a blood pressure of 90 mmHg), advanced cardiac treatment (coronary angiography, percutaneous coronary intervention, use of thrombolytics, extracorporeal bypass) and adverse events within 7 days to maintain a ROSC (seizure), and advanced cardiac treatment (coronary angiography).

According to the criteria of the World Health Organization (WHO), BMI was calculated as the weight in kilograms (kg) divided by height in meters squared. Based on the weight and height from the registry data, the calculated BMI was categorized as underweight (< 18.5 kg/m$^2$), normal weight (18.5–24.9 kg/m$^2$), overweight (BMI 25–29.9 kg/m$^2$) or obese ($\geq$ 30 kg/m$^2$).

## Primary outcome and subgroup analysis

The primary outcome was the neurologic outcome 6 months after resuscitation following OHCA. The outcomes were assessed by investigators at each center through face-to-face or telephone interviews. The neurologic outcome was expressed as good or poor; good performance (CPC 1) or moderate disability (CPC 2) were categorized as a good outcome; and severe disability (CPC 3), vegetative state (CPC 4) or death/brain death (CPC 5) were classified as a poor outcome. Subgroup analysis was performed to assess the effect of BMI on neurologic outcome in terms of patient characteristics and TTM variables.

## Statistical analysis

Recorded data were analyzed using IBM SPSS 24.0 for Windows (IBM corp., Armonk, NY, USA). Continuous variables are presented as the mean ± SD or median and interquartile range (IQR). Categorical variables are presented as counts (percentage). For comparisons among BMI groups, continuous variables were compared using the Kruskal-Wallis test when they did not follow a normal distribution according to the Kolmogorov-Smirnov test. The $\chi^2$ test was used for comparisons of categorical data. To determine the relationship between BMI and six-month neurologic outcome, univariate and multivariate analyses were performed with logistic regression using normal BMI as the reference category. The Hosmer-Lemeshow test was employed to examine goodness of fit, and odds ratios (ORs) and 95% confidence intervals (CIs) were computed. Subgroup analysis was conducted using forest plots and p-values to investigate the effect of various variables on the association between neurologic outcome and BMI. All statistical tests were two-tailed, and p-values < 0.05 indicated significant differences.

## Results

Of 1,373 patients identified in the KORHN-PRO registry, 24 with missing height and weight data and 34 without six-month neurologic outcomes in the database were excluded. Finally, 1,315 patients were enrolled and analyzed in this study. BMI was calculated and categorized into underweight, normal weight, overweight and obese groups according to the WHO criteria (Fig 1).

The baseline characteristics of the patients were analyzed by BMI category (Table 1). The majority of all BMI groups consisted of males (p < 0.001). Significant differences among the groups existed in age, hypertension, diabetes mellitus, lung disease, initial rhythm, arrest

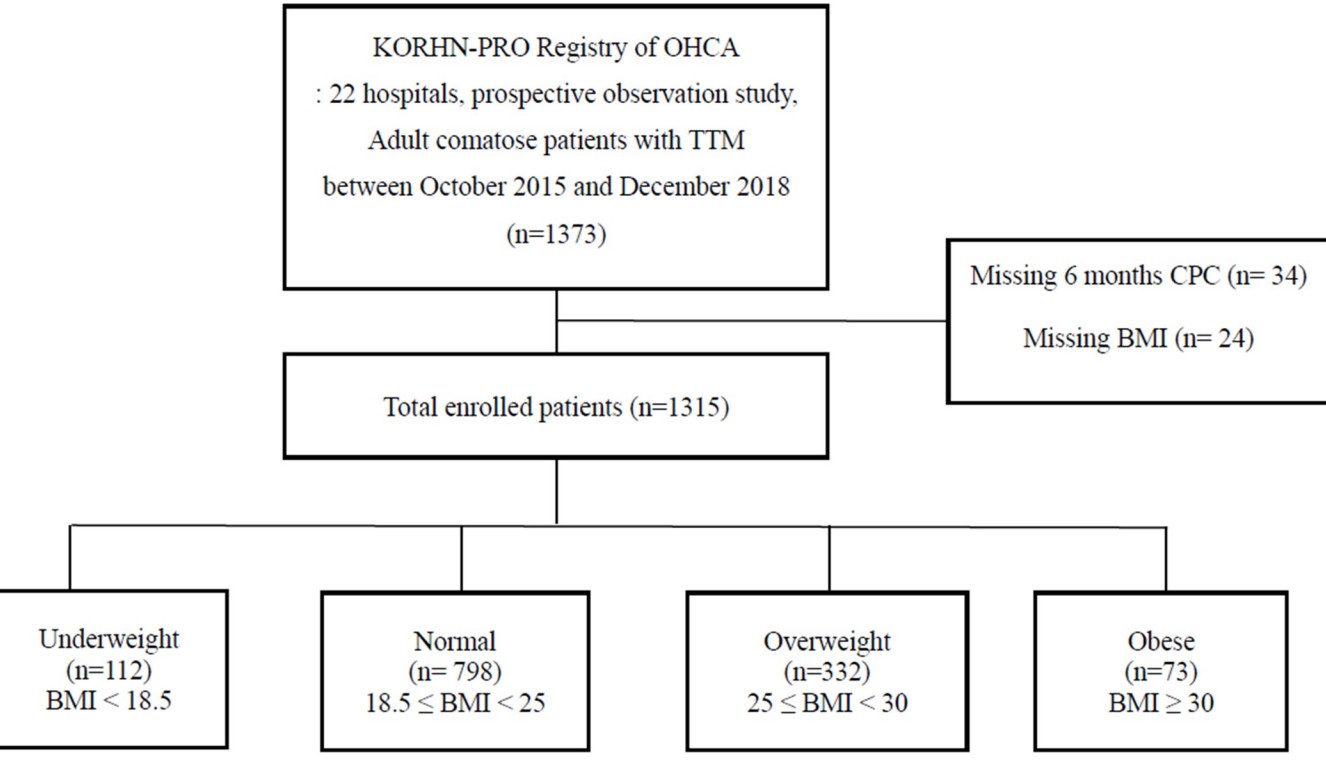

**Fig 1. Flow chart of the study.** *Abbreviation*: *OHCA* out-of-hospital cardiac arrest, *TTM* targeted temperature management, *BMI* body mass index.

**Table 1. Baseline characteristics of patients stratified according to body mass index classification.**

| | Total | Underweight | Normal | Overweight | Obese | p-value |
|---|---|---|---|---|---|---|
| | (n = 1315) | (n = 112) | (n = 798) | (n = 332) | (n = 73) | |
| Male sex | 933 (71.0%) | 65 (58.0%) | 579 (72.6%) | 249 (75.0%) | 40 (54.8%) | < 0.001 |
| Age, years, median (IQR) | 53.0 (48.0–70.0) | 60.0 (47.5–72.0) | 59.0 (48.0–70.0) | 57.0 (47.0–69.0) | 54.0 (44.0–63.0) | 0.042 |
| Comorbidities | | | | | | |
| CAD | 194 (14.8%) | 7 (6.3%) | 120 (15.0%) | 56 (16.9%) | 11 (15.1%) | 0.053 |
| Hypertension | 473 (36.0%) | 19 (17.0%) | 281 (35.2%) | 134 (40.4%) | 39 (53.4%) | < 0.001 |
| Diabetes mellitus | 318 (24.2%) | 15 (13.4%) | 192 (24.1%) | 84 (25.3%) | 27 (37.0%) | 0.003 |
| CVA | 73 (5.6%) | 8 (7.1%) | 47 (5.9%) | 15 (4.5%) | 3 (4.1%) | 0.641 |
| Lung disease | 84 (6.4%) | 22 (19.6%) | 41 (5.1%) | 17 (5.1%) | 4 (5.5%) | < 0.001 |
| Renal disease | 100 (7.6%) | 6 (5.4%) | 57 (7.1%) | 28 (8.4%) | 9 (12.3%) | 0.297 |
| BMI, kg/m$^2$, median (IQR) | 23.36 (20.96–25.71) | 17.30 (16.26–17.96) | 22.48 (20.76–23.76) | 26.49 (25.71–27.72) | 32.67 (31.06–35.23) | < 0.001 |
| Arrest location | | | | | | 0.366 |
| Home/residence | 674 (51.3%) | 64 (57.1%) | 410 (51.4%) | 160 (48.2%) | 40 (54.8%) | |
| Other | 641 (48.7%) | 48 (42.9%) | 388 (48.6%) | 172 (51.8%) | 33 (45.2%) | |
| Initial arrest rhythm | | | | | | 0.012 |
| Shockable | 443 (33.7%) | 26 (23.2%) | 270 (33.8%) | 128 (38.6%) | 19 (26.0%) | |
| Non-shockable | 872 (66.3%) | 86 (76.8%) | 528 (66.2%) | 204 (61.4%) | 54 (74.0%) | |
| Witnessed collapse | 910 (69.2%) | 74 (66.1%) | 534 (66.9%) | 247 (74.4%) | 55 (75.3%) | 0.047 |
| Bystander CPR | 810 (61.6%) | 66 (58.9%) | 486 (60.9%) | 210 (63.3%) | 48 (65.8%) | 0.701 |
| Arrest etiology | | | | | | 0.001 |
| Cardiac etiology | 812 (61.7%) | 52 (46.4%) | 492 (61.7%) | 226 (68.1%) | 42 (57.5%) | |
| Other etiology | 503 (38.3%) | 60 (53.6%) | 308 (38.3%) | 106 (31.9%) | 31 (42.5%) | |
| Anoxic time, minutes, median (IQR) | 3.0 (17.0–43.0) | 24.0 (13.0–38.0) | 28.0 (17.0–40.0) | 28.0 (17.0–43.0) | 32.0 (15.0–45.5) | 0.047 |
| TTM methods | | | | | | 0.756 |
| Surface cooling | 1178 (89.6%) | 99 (88.4%) | 717 (89.8%) | 296 (89.2%) | 66 (90.4%) | |
| Intravascular cooling | 122 (9.3%) | 13 (11.6%) | 71 (8.9%) | 31 (9.3%) | 7 (9.6%) | |
| Other | 15 (1.1%) | 0 (0.0%) | 10 (1.3%) | 5 (1.5%) | 0 (0.0%) | |
| TTM target temperature | | | | | | **0.275** |
| 33°C | 1035 (78.7%) | 85 (75.9%) | 618 (77.4%) | 272 (81.9%) | 60 (82.2%) | |
| 36°C | 280 (21.3%) | 27 (24.1%) | 180 (22.6%) | 60 (18.1%) | 13 (17.8%) | |
| Induction time, hours, median (IQR) | 2.33 (1.08–4.25) | 1.42 (1.00–3.50) | 2.25 (1.07–4.19) | 3.00 (1.83–4.92) | 3.75 (1.58–5.88) | < 0.001 |
| Rewarming time, hours, median (IQR) | 13.67 (9.27–16.00) | 13.00 (8.63–17.80) | 13.5 (9.00–16.00) | 14.00 (10.00–16.00) | 14.00 (8.50–16.00) | 0.226 |
| ROSC to TTM, hours, median (IQR) | 3.42 (2.17–4.87) | 3.62 (2.18–7.21) | 3.45 (2.17–5.06) | 3.33 (2.20–4.73) | 4.00 (2.83–5.13) | 0.466 |
| ROSC to target, hours, median (IQR) | 6.20 (4.28–9.00) | 5.42 (3.89–7.38) | 6.22 (4.26–9.00) | 6.83 (4.88–9.72) | 7.13 (4.93–10.10) | < 0.001 |
| Shock after ROSC | 638 (48.5%) | 41 (36.6%) | 396 (49.8%) | 164 (49.5%) | 37 (50.7%) | 0.067 |
| Reperfusion | | | | | | < 0.001 |
| None | 831 (63.2%) | 91 (81.3%) | 495 (62.0%) | 204 (61.4%) | 41 (56.2%) | |
| CAG only | 285 (21.7%) | 13 (11.6%) | 191 (23.9%) | 62 (18.7%) | 19 (26.0%) | |
| CAG with PCI | 199 (15.1%) | 8 (7.1%) | 112 (14.0%) | 66 (19.9%) | 13 (17.8%) | |
| tPA | 40 (3.0%) | 3 (2.7%) | 24 (3.0%) | 12 (3.6%) | 1 (1.4%) | 0.774 |
| ECMO | 59 (4.5%) | 2 (1.8%) | 35 (4.4%) | 18 (5.4%) | 4 (5.5%) | 0.428 |
| Adverse events | | | | | | |
| Seizure | 316 (24.0%) | 39 (34.8%) | 194 (24.3%) | 66 (19.9%) | 17 (23.3%) | 0.016 |
| Bleeding | 64 (4.9%) | 5 (4.5%) | 40 (5.0%) | 17 (5.1%) | 2 (2.7%) | 0.96 |
| Pneumonia | 512 (38.9%) | 55 (49.1%) | 304 (38.1%) | 127 (38.3%) | 26 (35.6%) | 0.138 |
| Sepsis | 153 (11.6%) | 15 (13.4%) | 89 (11.2%) | 39 (11.7%) | 10 (13.7%) | 0.844 |
| Rearrest | 248 (18.9%) | 15 (13.4%) | 140 (17.5%) | 74 (22.3%) | 19 (26.0%) | 0.044 |

*(Continued)*

**Table 1.** (Continued)

|  | Total | Underweight | Normal | Overweight | Obese | p-value |
|---|---|---|---|---|---|---|
|  | (n = 1315) | (n = 112) | (n = 798) | (n = 332) | (n = 73) |  |
| Six-month neurologic outcome |  |  |  |  |  | 0.111 |
| Good outcome | 410 (31.2%) | 25 (22.3%) | 262 (32.8%) | 104 (31.3%) | 19 (26.0%) |  |
| Poor outcome | 905 (68.8%) | 87 (77.7%) | 536 (67.2%) | 228 (68.7%) | 54 (74.0%) |  |
| Six-month mortality | 541 (41.1%) | 39 (34.8%) | 342 (42.9%) | 135 (40.8%) | 25 (34.2%) | 0.234 |

*Abbreviations*: *IQR* interquartile range, *CAD* coronary artery disease, *CVA* cerebrovascular accident, *BMI* body mass index, *CPR* cardiopulmonary resuscitation, *TTM* targeted temperature management, ROSC return of spontaneous circulation, CAG coronary angiography, PCI percutaneous coronary intervention, tPA tissue plasminogen activator, *ECMO* extracorporeal membrane oxygenation.

Continuous variables are presented as the median (Q1-Q3) and tested by using the Kruskal-Wallis test, and categorical variables are presented as N (%) and tested by using the chi-squared test.

*$p < 0.05$ was significant.

etiology, total anoxic time, induction time, ROSC to target temperature and reperfusion treatment. In particular, older patients had lower BMIs ($p < 0.001$), and patients with higher BMIs had greater incidences of hypertension ($p < 0.001$), diabetes mellitus ($p = 0.003$) and witnessed collapse ($p = 0.047$) and a longer induction time ($p < 0.001$) and time from ROSC to the target temperature ($p < 0.001$).

The primary neurologic outcomes were compared by BMI classification. Of the 1,315 included patients, 6 months after cardiac arrest, 905 patients had poor neurologic outcomes, and 541 patients had not survived. The number and frequency of patients in each BMI group are presented in Table 1. The results showed no significant difference among the groups.

Univariate and multivariate logistic regression analyses were performed to identify the relationship between BMI and the primary outcome (Table 2). In the univariate analysis, underweight patients were more frequently associated with poor neurologic outcomes (odds ratio [OR], 1.701; 95% confidence interval [CI], 1.065–2.718, $p = 0.026$). Both univariate and multivariate analyses revealed significant associations with age, initial arrest rhythm, arrest etiology, anoxic time and shock after ROSC. However, in the multivariate logistic regression analysis, we found that BMI was not associated with poor neurologic outcome after adjusting for sex, age, history of hypertension, diabetes mellitus, lung disease, renal disease, arrest location, initial arrest rhythm, witnessed collapse, bystander CPR, arrest etiology, anoxic time, shock after ROSC.

Subgroup analysis showed that BMI had no effect on neurologic outcomes except that underweight patients treated with a target temperature of 33˚C tended to have poor neurologic outcomes (odds ratio [OR] 2.090; 95% confidence interval [CI] 1.010–4.325; $p = 0.047$) (Table 3, Fig 2).

## Discussion

The purpose of this study was to investigate the influence of BMI on the neurologic outcome of OHCA survivors treated with TTM. Because obesity is a risk factor for cardiovascular disease and death [2, 3], obese and overweight patients who survive CA are assumed to have a worse prognosis. When comparing characteristics among the BMI groups, our data showed that patients with higher BMIs had a greater incidence of hypertension and diabetes mellitus. Additionally, obese and overweight patients required more time to reach the target temperature. Although these differences existed, neurologic outcome was not associated with BMI in the univariate and multivariate logistic regression analyses. Only in the subgroup analysis

**Table 2. Univariate and multivariate logistic regression analysis for poor neurologic outcome at 6 months.**

| Variables | Univariate analysis | | | Multivariate analysis | | |
|---|---|---|---|---|---|---|
| | Unadjusted OR | 95% CI for the OR | p-value | Adjusted OR | 95% CI for the OR | p-value |
| Sex (Male/Female) | 0.602 | 0.459–0.790 | < 0.001 | 0.778 | 0.535–1.131 | 0.189 |
| Age (per years) | 1.032 | 1.024–1.039 | < 0.001 | 1.044 | 1.031–1.057 | < 0.001 |
| Comorbidities | | | | | | |
| Coronary heart disease | 1.263 | 0.917–1.739 | 0.154 | | | |
| Hypertension | 1.545 | 1.202–1.987 | 0.001 | 1.037 | 0.705–1.524 | 0.853 |
| Diabetes mellitus | 2.109 | 1.557–2.857 | < 0.001 | 1.299 | 0.838–2.013 | 0.241 |
| CVA | 1.409 | 0.817–2.431 | 0.218 | | | |
| Lung disease | 4.026 | 1.996–8.121 | < 0.001 | 0.971 | 0.425–2.218 | 0.945 |
| Renal disease | 2.334 | 1.367–3.987 | 0.002 | 1.242 | 0.625–2.466 | 0.536 |
| Arrest location (Home/Others) | 1.674 | 1.323–2.119 | < 0.001 | 1.092 | 0.784–1.520 | 0.603 |
| Initial arrest rhythm (Shockable/Non-shockable) | 0.089 | 0.068–0.117 | < 0.001 | 0.183 | 0.126–0.264 | < 0.001 |
| Witnessed collapse | 0.32 | 0.238–0.431 | < 0.001 | 0.687 | 0.460–1.026 | 0.066 |
| Bystander CPR | 0.677 | 0.530–0.866 | 0.002 | 1.099 | 0.780–1.550 | 0.589 |
| Cardiac arrest etiology | 0.135 | 0.097–0.187 | < 0.001 | 0.232 | 0.148–0.364 | < 0.001 |
| Time from arrest to ROSC (per minutes) | 1.061 | 1.051–1.071 | < 0.001 | 1.064 | 1.052–1.076 | < 0.001 |
| TTM target temperature (33˚C/36˚C) | 0.965 | 0.726–1.282 | 0.805 | | | |
| ROSC to TTM start time (per hours) | 0.98 | 0.940–1.021 | 0.332 | | | |
| ROSC to target temperature time (per hours) | 1 | 0.999–1.001 | 0.513 | | | |
| Shock after ROSC (Yes/No) | 2.819 | 2.209–3.597 | < 0.001 | 1.888 | 1.362–2.618 | < 0.001 |
| BMI, kg/m$^2$ | | | | | | |
| Normal (15.5–24.9) | Reference | - | - | | | |
| Underweight (< 18.5) | 1.701 | 1.065–2.718 | 0.026 | 1.668 | 0.903–3.080 | 0.102 |
| Overweight (25.0–29.9) | 1.072 | 0.814–1.411 | 0.622 | 1.444 | 0.981–2.125 | 0.062 |
| Obese (> 30.0) | 1.389 | 0.807–2.392 | 0.236 | 1.421 | 0.684–2.951 | 0.347 |

*Abbreviations*: *OR* odds ratio, *CI* confidence interval, *CVA* Cerebrovascular accident, *CPR* cardiopulmonary resuscitation, *TTM* targeted temperature management, ROSC return of spontaneous circulation, *BMI* body mass index.

*$p<0.05$ was significant.

**Table 3. Comparison of odds ratios by body mass index group for poor neurologic outcomes based on the target temperature.**

| | | OR | 95% CI for the OR | p-value |
|---|---|---|---|---|
| 33˚C | Underweight | 2.09 | 1.010–4.325 | 0.047 |
| | Overweight | 1.399 | 0.908–2.156 | 0.128 |
| | Obese | 1.964 | 0.827–4.663 | 0.126 |
| 36˚C | Underweight | 0.88 | 0.249–3.112 | 0.843 |
| | Overweight | 1.641 | 0.653–4.123 | 0.292 |
| | Obese | 0.519 | 0.100–2.704 | 0.436 |

*Abbreviations*: *OR* odds ratio, *CI* confidence interval.

*Adjusted sex, age, HTN, DM, lung disease, renal disease, arrest location, initial rhythm, witnessed arrest, bystander CPR, cardiac etiology, TTM shock.

*$p<0.05$ was significant.

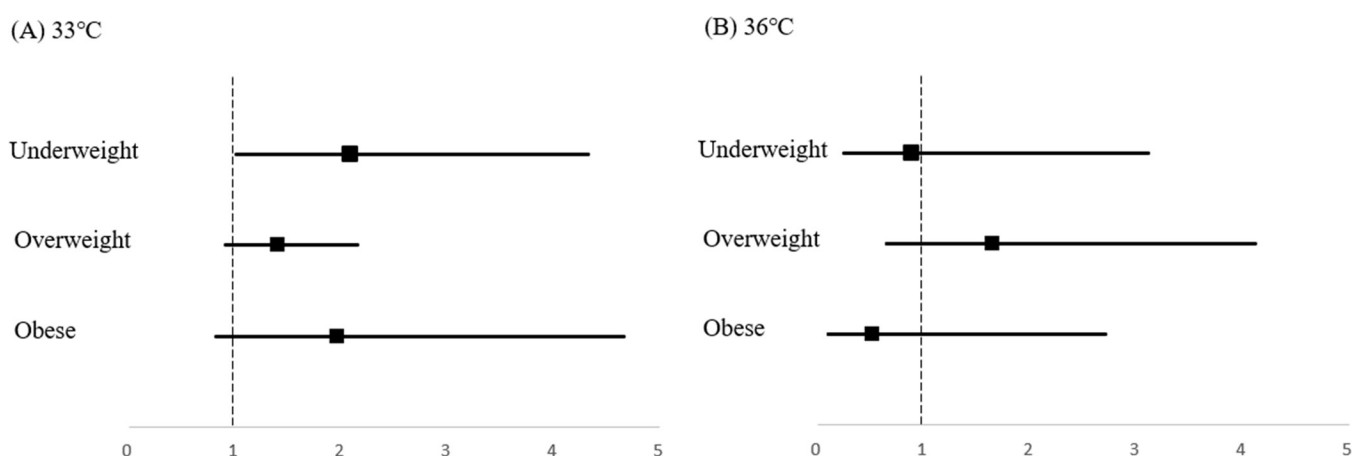

**Fig 2. Forest plots of the odds ratios among the BMI groups according to the target temperature.** (A) The odds ratio of 33˚C targeted temperature management (B) The odds ratio of 36˚C targeted temperature management.

between those treated with target temperatures of 33˚C and 36˚C was underweight status and treatment at 33˚C associated with poor neurologic outcomes.

The association between BMI and OHCA outcomes has been the subject of several studies. Our study found that BMI was not associated with poor neurologic outcomes, although higher BMIs increased the incidence of underlying disease and cardiovascular death. More than half of the enrolled patients were initially found to have a nonshockable rhythm, asystole in particular, which may cause poor outcomes in CA [13]. Moreover, the anoxic time from collapse to ROSC was a significant factor in the neurologic outcome of patients who experienced CA [14]. In the KORHN-PRO registry, the median anoxic time was 30 minutes, which contributed considerably to unfavorable neurologic outcomes. In particular, patients in the obese group had more nonshockable rhythms at arrest and much longer total anoxic times than those in the other groups, while witnessed CA was significantly more common in the obese group. Since the relative importance of BMI and other cardiac arrest factors could not be weighted, the effect of BMI on neurologic outcomes might be affected. One possible explanation for this result is that the ischemia-reperfusion process after ROSC leads to systemic inflammatory injury, called sepsis-like syndrome, and affects patient outcomes [15]. In a previous study on sepsis and BMI, adipose tissue was shown to regulate immunity by secreting anti-inflammatory adipokines and providing fuel in acute life-threatening illness [16, 17]. In this registry, there were some demographic factors that could adversely affect the prognosis of obese patients, but for this reason, the neurological prognosis after 6 months may not be significantly different from that of other groups.

Previous studies demonstrated that a higher BMI was related to a delay in induction time for TTM. Yong Hun Jung et al. showed a prolonged induction duration and slower cooling rate in the obese group [18]. These findings were supported by adipose tissue functioning as an insulator and heat trap [19]. Therefore, obese and overweight patients require a relatively extended time to reach the target temperature, and more aggressive and complex cooling methods might be necessary.

In the subgroup analysis, TTM at 33˚C in underweight patients with OHCA resulted in unfavorable neurologic outcomes. This might be explained by underweight patients being sensitive to low temperature and in turn achieving poor outcomes. A prior study highlighted that patients with lower BMIs suffered from more cold injury and peripheral vascular disease than patients with higher BMIs due to a shortage of fat tissue [20]. Furthermore, the results were

correlated with the obesity paradox and better outcomes with higher BMIs [21]. In a meta-analysis by Ma, Y., 24,822 patients from 7 studies were analyzed to clarify the relationship between BMI and the clinical outcomes of patients who experienced CA. The study found that overweight patients had a favorable neurological prognosis after CA [22]. Gupta stated that obese patients had higher risk-adjusted odds of survival not only in OCHA but also in IHCA (in-hospital cardiac arrest) [23]. These phenomena resulted from adipose tissue providing a metabolic reserve and lipid soluble nutrients during a highly active metabolic state, such as CA and low temperature during TTM [24]. Consequently, when considering TTM at 33˚C for underweight patients resuscitated from OHCA, clinicians should pay special attention to their treatment approach to achieve favorable outcomes.

One of the strengths of this study was that the KORHN-PRO registry contained a large number of OHCA patients from multiple centers. In addition, few patients decided to withdraw life support treatment, and the risk of self-fulfilling prophecy could be avoided. Another strength of this study was that all enrolled patients were provided with well-organized TTM and intensive care. This alleviated treatment gaps among hospitals. Consequently, the body temperature of patients was maintained at a constant temperature, and fluctuation was minimized during TTM. To the best of our knowledge, the current study is the first registry study comparing TTM 33˚C and 36˚C treatments among BMI groups.

This study had several limitations. First, our study is an observational prospective study, and there is a risk of selection bias and residual confounding. Second, even though the TTM protocol was uniform, clinicians set the target temperature of TTM based on individual preferences, which may influence outcomes. Third, only a few obese patients were included, so the effect of the obese group might be underestimated because the number of Asians with a BMI above 30 was small. Fourth, we used BMI as an indicator of underweight and obesity. Because BMI does not consider body fat distribution or distinguish lean body mass from fat mass, supplementary values could have been applied [25].

## Conclusions

By examining the KORHN-PRO registry, we found that BMI was not an independent risk factor for poor neurologic outcomes at 6 months in CA survivors treated with TTM. In addition, aggressive and additional cooling methods should be considered for patients with higher BMIs, and special attention might be needed for underweight patients in the TTM 33 group.

## Supporting information

**S1 Data.**
(PDF)

## Acknowledgments

The following investigators participated in the Korean Hypothermia Network.Network chair: Seung Pill Choi (The Catholic University of Korea, Eunpyeong St. Mary's Hospital);principal investigators of each hospital: Kyu Nam Park (The Catholic University of Korea, Seoul St. Mary's Hospital), Minjung Kathy Chae (Ajou University Medical Center), Won Young Kim (AsanMedical Center), Byung Kook Lee (Chonnam National University Hospital), Dong Hoon Lee (Chung-Ang University Hospital), Tae Chang Jang (Daegu Catholic University Medical Center), Jae Hoon Lee (Dong-A University Hospital), Yoon Hee Choi (Ewha Womans University Mokdong Hospital),Je Sung You (Gangnam Severance Hospital), Young Hwan Lee (Hallym University Sacred HeartHospital), In Soo Cho (Hanil General Hospital), Su Jin

Kim (Korea University Anam Hospital), Jong-Seok Lee (Kyung Hee University Medical Center), Yong Hwan Kim (Samsung Changwon Hospital),Min Seob Sim (Samsung Medical Center), Jonghwan Shin (Seoul Metropolitan Government SeoulNational University Boramae Medical Center), Yoo Seok Park (Severance Hospital), Hyung Jun Moon(Soonchunhyang University Hospital Cheonan), Won Jung Jeong (The Catholic University of Korea, St. Vincent's Hospital), Joo Suk Oh (The Catholic University of Korea, Uijeongbu St. Mary's Hospital),Seung Pill Choi (The Catholic University of Korea, Yeouido St. Mary's Hospital), Kyoung-Chul Cha(Wonju Severance Christian Hospital).

## Author Contributions

**Conceptualization:** Hyo Jin Bang, Kyu Nam Park, Hyo Joon Kim.

**Data curation:** Jee Yong Lim, Won Jung Jeong.

**Formal analysis:** Hwan Song, Soo Hyun Kim.

**Investigation:** Hyo Jin Bang, Han Joon Kim, Hyo Joon Kim.

**Methodology:** Hyo Joon Kim.

**Supervision:** Chun Song Youn, Sang Hoon Oh.

**Writing – original draft:** Hyo Jin Bang.

**Writing – review & editing:** Hyo Joon Kim.

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
