## [Decision Letter · Decision Letter 0]

19 Jan 2022

PONE-D-21-35740The Relationship Between Body Mass Index and Neurologic Outcomes in Survivors of Out-of-Hospital Cardiac Arrest Treated with Targeted Temperature ManagementPLOS ONE

Dear Dr. Kim,

Thank you for submitting your manuscript to PLOS ONE. After careful consideration, we feel that it has merit but does not fully meet PLOS ONE’s publication criteria as it currently stands. Therefore, we invite you to submit a revised version of the manuscript that addresses the points raised during the review process.

The authors concluded that BMI is not associated with neurologic outcomes in out of hospital cardiac arrest treated with TTM. Can the authors report what were the variables used in multivariables analysis?

We look forward to receiving your revised manuscript.

Kind regards,

Jignesh K. Patel

Academic Editor

PLOS ONE

Journal Requirements:

2. Please include the full name of the IRB or ethics committee who approved or waived your study

Reviewers' comments:

Reviewer's Responses to Questions

**Comments to the Author**

1. Is the manuscript technically sound, and do the data support the conclusions?

Reviewer #1: Yes

2. Has the statistical analysis been performed appropriately and rigorously? 

Reviewer #1: Yes

3. Have the authors made all data underlying the findings in their manuscript fully available?

Reviewer #1: Yes

4. Is the manuscript presented in an intelligible fashion and written in standard English?

Reviewer #1: Yes

5. Review Comments to the Author

Reviewer #1: A very interesting paper that helps argue against the thinking that higher BMI is always correlated with worse outcomes. I appreciated the discussion section that tried to provide reasons behind the observed findings in the study.

There seems to be a formatting issue with the manuscript and I was unable to view the "Obese" column for Table 1 and multivariate analysis in Table 2.

6. PLOS authors have the option to publish the peer review history of their article (what does this mean?). If published, this will include your full peer review and any attached files.

Reviewer #1: No

---

## [Author Response · Author response to Decision Letter 0]

17 Feb 2022

Editors’ comments: 

1) Can the authors report what were the variables used in multivariables analysis?

Thank you for your comments. I agree with your comment and I added variables in mnauscripts.

However, in the multivariate logistic regression analysis, we found that BMI was not associated with poor neurologic outcome after adjusting for sex, age, history of hypertension, diabetes mellitus, lung disease, renal disease, arrest location, initial arrest rhythm, witnessed collapse, bystander CPR, arrest etiology, anoxic time, shock after ROSC.

2) Please include the full name of the IRB or ethics committee who approved or waived your study.

 Thank you for your comments. I added IRB in Manuscripts.

The study included an informed consent form approved by all participating hospitals, including the institutional review board (IRB) of Seoul St. Mary’s Hospital (XC150IMI0081K)

Reviewer #1

A very interesting paper that helps argue against the thinking that higher BMI is always correlated with worse outcomes. I appreciated the discussion section that tried to provide reasons behind the observed findings in the study.

1) There seems to be a formatting issue with the manuscript and I was unable to view the "Obese" column for Table 1 and multivariate analysis in Table 2.

Thank you for your comments. I changed Tables 1 and Table 2 that can view “Obeses” columns.

---

## [Editor Report · Decision Letter 1]

7 Mar 2022

The Relationship Between Body Mass Index and Neurologic Outcomes in Survivors of Out-of-Hospital Cardiac Arrest Treated with Targeted Temperature Management

PONE-D-21-35740R1

Dear Dr. Kim,

We’re pleased to inform you that your manuscript has been judged scientifically suitable for publication and will be formally accepted for publication once it meets all outstanding technical requirements.

Kind regards,

Jignesh K. Patel

Academic Editor

PLOS ONE

Additional Editor Comments (optional):

The author's have appropriately answered all the questions.
---

## [Editor Report · Acceptance letter]

16 Mar 2022

PONE-D-21-35740R1 

The Relationship Between Body Mass Index and Neurologic Outcomes in Survivors of Out-of-Hospital Cardiac Arrest Treated with Targeted Temperature Management 

Dear Dr. Kim:

I'm pleased to inform you that your manuscript has been deemed suitable for publication in PLOS ONE. Congratulations! Your manuscript is now with our production department. 

Kind regards, 

on behalf of

Dr. Jignesh K. Patel 

Academic Editor

PLOS ONE